# Effect of Partial Meat Replacement by *Hibiscus sabdariffa* By-Product and *Pleurotus djamor* Powder on the Quality of Beef Patties

**DOI:** 10.3390/foods12020391

**Published:** 2023-01-13

**Authors:** Roberto Bermúdez, Esmeralda Rangel-Vargas, José M. Lorenzo, José A. Rodríguez, Paulo E. S. Munekata, Alfredo Teixeira, Mirian Pateiro, Leticia Romero, Eva M. Santos

**Affiliations:** 1Centro Tecnológico de la Carne de Galicia, Avd. Galicia No. 4, Parque Tecnológico de Galicia, San Cibrao das Viñas, 32900 Ourense, Spain; 2Área Académica de Química, Universidad Autónoma del Estado de Hidalgo, Crta. Pachuca-Tulancingo Km 4.5 s/n, Col. Carboneras, Mineral de la Reforma 42183, Mexico; 3Área de Tecnoloxía dos Alimentos, Facultade de Ciencias, Universidade de Vigo, 32004 Ourense, Spain; 4Centro de Investigação de Montanha (CIMO), Instituto Politécnico de Bragança, Campus de Santa Apolónia, 5300-253 Bragança, Portugal; 5Área Académica de Biología, Universidad Autónoma del Estado de Hidalgo, Crta. Pachuca-Tulancingo Km 4.5 s/n, Col. Carboneras, Mineral de la Reforma 42183, Mexico

**Keywords:** roselle, pink oyster, edible mushroom, healthy meat products, antimicrobial, antioxidant, meat replacement, plant-based foods

## Abstract

The effects of *Hibiscus sabdariffa* (roselle; Hs) by-product (2–5%) and *Pleurotus djamor* (pink oyster; Pd) powder (5–7.5%) as meat replacers on the physicochemical and sensorial properties of beef patties were analyzed. The addition of these non-meat ingredients significantly decreased moisture and increased fiber content, and did not affect the protein level of the product. The antioxidant effect of the roselle by-product was limited, while *Pleurotus djamor* favored the oxidation processes. The samples supplemented with roselle by-product and mushroom powder presented significantly lower microbial counts (total viable counts, enterobacteria, and *Pseudomonas*) than control, but texture and sensorial parameters were significantly affected. The patties darkened due to the presence of the *Hibiscus* by-product, while the color of the samples containing 5% *Pleurotus djamor* was hardly modified. These samples, together with the control samples, were the most sensorially appreciated. The addition of these ingredients provoked a decrease in texture parameters, being less pronounced in the samples with only 2% of roselle by-product. In spite of the good antimicrobial and antioxidant properties of *Hibiscus* by-products, its inclusion in meat should be moderate (2–2.5%) to avoid consumer rejection when *Pleurotus djamor* is also included in the formulation.

## 1. Introduction

The recent concerns about the sustainability of meat production chains, as well as the negative health issues associated with animal-based diets (such as hypertension, cardiovascular diseases, obesity, metabolic syndrome, and even cancer) have resulted in an intensification of research activities to develop healthier meat products [1]. The main trend is the use of vegetable sources or even by-products to substitute meat, fat, or other unbeneficial components in an attempt to improve the nutritional profile of meat products and offer functional properties, which can exert a beneficial effect on consumers’ health [2]. In this regard, there is abundant literature about the use of different plant-based ingredients and alternative proteins used in meat products [3,4].

Recently, edible mushrooms have received attention as one of the main alternative ingredients for the nutritional improvement of meat products [5]. The noteworthy components of edible mushrooms supporting this interest are their high protein and dietary fiber content, as well as their characteristic umami taste, which can be used to improve the sensory and nutritional qualities of meat products [6,7]. However, the acceptability of the reformulated meat products can be highly influenced by how familiarized the consumer is with the unique aroma and taste of these species [7,8].

Commonly cultivated species such as *Agaricus*, *Pleurotus* or those from the *Lentinula* genera impart a noticeable odor and flavor to meat products, which could be unwelcome for consumers [7]. The genus *Pleurotus* spp. is one of the best known because of its nutritional importance and the biofunctional properties it exhibits [6,9]. *Pleurotus djamor* (Pd), or pink oyster mushroom, is a lesser-known species from *Pleurotus* with a remarkable characteristic light-to-dark-pink color and a milder scent [10], which are favorable characteristics that indicate this mushroom to be an excellent candidate to develop meat products.

On the other hand, *Hibiscus sabdariffa* (Hs), also known as roselle, is another noteworthy vegetable because of its interesting antimicrobial and antioxidant properties and eye-catching red color [11]. Apart from the traditional use of roselle calyxes in the preparation of beverages or folk medicine (involving a decoction process), roselle has received a lot of attention as a natural source of compounds with functional properties [12], and its inclusion in foods (mainly as water extracts) has been studied in several works [11]. Although the decoction process extracts high-added value compounds from roselle, several functional components retained in the solid residue are discarded after the decoction [13]. Moreover, information about the use of roselle calyxes by-products after the decoction process as a food ingredient is scarce. 

When *Agaricus bisporus* and *Pleurotus ostreatus* were added to beef patties in a previous work [7], despite the improvement of protein and fiber contents, no antioxidant effects were observed because of the addition of these compounds. Considering the nutritional value of pink oyster mushrooms and the potential antioxidant properties of *Hibiscus* by-products, the aim of this study was to analyze the effect of the partial meat replacement by *Pleurotus djamor* and *Hibiscus* by-product powders on the physicochemical, microbiological, and sensorial properties of beef patties.

## 2. Materials and Methods

### 2.1. Preparation and Characterization of Pd and Hs Ingredients

*Pleurotus djamor* (Pd) species was provided by Biology Academic Area, Universidad Autónoma del Estado de Hidalgo (Mexico). The mushrooms were collected, rinsed with clean water, drained, and dried at 40 °C for 2–3 days in an industrial oven. In the case of *Hibiscus sabdariffa* (Hs) by-products, the material used was the solid residue generated from the extraction of antioxidants from ground *Hibiscus sabdariffa calyxes* in water (1:10) after centrifugation at 1109× *g* for 15 min [14]. The solid residues from centrifugation were dried in the same conditions applied to the mushrooms. After that, the samples were milled in a grinder (UDI Samples Millis, Fort Collins, CO, USA) to a particle size smaller than 0.5 mm, packaged in hermetic polyethylene bags, and stored in darkness until use. Chemical composition of Pd and Hs powders was determined as described in the following sections.

### 2.2. Antioxidant Activity and Polyphenol Determination of Pd and Hs Powders

Methanolic extracts were obtained with 100 mg of Pd or Hd powder in 10 mL of MeOH. The mixtures were sonicated for 10 min. Then, the extracts were centrifuged at 596× *g* and filtered through filter paper (Filter-lab 1238, Barcelona, Spain). The antioxidant activity of methanolic extracts was characterized by DPPH, ABTS, and FRAP assays following the protocol described by Rivero-Perez et al. [15]. The results were expressed as mg Trolox/g sample in the case of DPPH and ABTS assays. The FRAP method was performed according to the method described by Benzie et al. [16], and results were expressed in mmol FeSO_4_/100 g sample. Additionally, total polyphenol content was determined by the Folin–Ciocalteu reagent [15], using gallic acid as standard and expressing the results as mg gallic acid equivalents (GAE)/g sample.

### 2.3. Beef Patties Manufacture

The burgers were produced in the pilot plant of the *Centro Tecnológico de la Carne* (Ourense, Spain). Beef patties were produced with thawed meat (under refrigeration since the night before the processing of patties). Beef meat was minced with a plate of 8 mm in a refrigerated chopper mill (La Minerva, Bologna, Italy). The ground meat was divided into 6 treatments (2 kg each): control, Pd5, Hs2, Hs5, Pd5-Hs2, and Pd7.5-Hs2.5. The Pd powder replaced 5% of meat in formulations Pd5 and Pd5-Hs2, and 7.5% of meat in the formulation coded as Pd7.5-Hs2.5 (Figure 1). The Hs powder replaced 2% of meat in two formulations (Hs2 and Pd5-Hs2), 2.5% in formulation Pd7.5-Hs2.5, and 5% in the formulation coded as Hs5. All treatments were processed with low NaCl addition (0.7%) and 10% of cold water to facilitate the incorporation of Pd and Hs powders in a mixer (RM-20, Mainca, Barcelona, Spain). Then, 80 g portions were molded into patties of 10 cm diameter and 1 cm height in a burger maker (A-2000, Gaser, Girona, Spain). Patties (2 per tray) were packaged in 300 mm thick PET-EVOH-PE trays with a gas mixture (80% O_2_/ 20% CO_2_) and sealed with multilayer PE-EVOH-PE film (74 mm thick, permeability <2 mL/m^2^ bar/day (Viduca, Alicante, Spain) using a heat sealer (LARI3/Pn T-VG-R-SKIN, Ca.Ve.Co., Palazzolo, Italy). Samples were stored at 4 ± 1 °C and four units of patties from each treatment were taken at 1, 4, 7, and 12 days of storage. After sampling for microbial analysis, the patties were left oxygenated for 30 min at room temperature. All tests were performed in duplicate.

### 2.4. Chemical Composition and Na Determination

Chemical composition analysis and Na content were determined only on day 1. Moisture, ash, and protein were quantified according to ISO recommended standards (1442:1997, 936:1998, and 937:1978, respectively) following the procedures described by Bermúdez et al. [17]. The fat was extracted in the fat extractor Ankom XT10 (ANKOM Technology Corp., Macedon, NY, USA) according to the A.O.C.S. Official Procedure Am 5–04 [17]. The carbohydrate content was calculated by the difference of the aforementioned components (100—[moisture (%)—protein (%)—fat (%)—ash (%)]) [18]. Total dietary fiber content was determined with the dietary fiber assay kit TDF-100 A (Sigma Aldrich, St. Louis, MI, USA) according to the AOAC Procedure [19].

The quantification of Na content was performed in ashes obtained from 5 g samples, previously incinerated, and dissolved in 10 mL of 1 M HNO_3_ according to the method described by Cutillas et al. [20]. The filtered samples were measured by inductively coupled plasma-optical emission spectroscopy (ICP-OES), using a Perkin-Elmer Avio 200 plasma emission spectrometer (Perkin Elmer, MA, USA) equipped with a radio frequency source of 27.12 MHz, a peristaltic pump, a spraying chamber, and a concentric spray nebulizer. The operating wavelength for Na quantification was 589.592 nm. The results were expressed in mg/100 g of sample.

### 2.5. TBARs Determination

Lipid oxidation was determined using the protocol described by Tarladgis et al. [21], which has a distillation stage as a preparatory stage to recover malonaldehyde (MDA) and prevent the interference of colored ingredients added to the meat (Pd, and specially Hs). Briefly, 10 g of the sample previously dispersed in distilled water with a homogenizer (IKA T-25 Basic, Staufen, Germany), acidified with 4 N HCL solution, and distilled to recover MDA. Five mL from the distillate were mixed with five mL of thiobarbituric acid (0.02 M in acetic acid at 90%) and the mixture was incubated at 96 °C for 40 min. The TBARs values, expressed as mg of MDA/kg sample, was calculated from the absorbance of the sample at 538 nm multiplied by a conversion factor of 7.8.

### 2.6. pH and Microbial Analyses

The pH values were obtained using a digital pH meter (HI 99163, Hanna Instruments, Eibar, Spain) equipped with a penetration glass probe. Microbiological analyses were performed in duplicate on each sampling day. Total Viable Counts (TVC), lactic acid bacteria, enterobacteria, and molds and yeast were determined with the TEMPO system, an automated enumeration system including a TEMPO Filler and a TEMPO Reader equipment (BioMérieux, Marcy l Étoile, France), based on the most likely number technique (detection limit of 0.3 CFU/g sample). Inoculated media for TVC and LAB micro-organisms were incubated at 30 °C for 24 and 48 h, respectively; while inoculated media for enterobacteria enumeration were incubated at 37 °C for 24 h. Incubation for mold/yeast analysis was carried out at 25 °C for 72 h. *Pseudomonas* spp. was determined by the classic technique on Pseudomonas Agar Base (Oxoid, Unipath Ltd., Basingstoke, UK) supplemented with CFC selective supplement (Oxoid, Basingstoke, UK) after incubation at 25 °C for 48 h. Psychrotrophic aerobic bacteria were enumerated on Plate Count Agar (PCA; Oxoid, Unipath Ltd., Basingstoke, UK) after incubation at 7 °C for 7–10 days. Counts were expressed in Log CFU/g.

### 2.7. Cooking Loss

This parameter was evaluated according to Heck et al. [22]. For each treatment, 30 g of sample were packaged and cooked in a water bath (at 80 °C) until reaching the internal temperature of 72 °C. After that, samples were cooled to room temperature. Cooking loss was calculated as weight difference (%) between cooked and fresh patties.

### 2.8. Color and Texture Evaluation

Color was measured at three different points on the surface of the samples using a portable colorimeter (Konica Minolta CM-600 d, Osaka, Japan) with an illuminant D65, 0° viewing angle geometry and 8 mm of aperture size. The color data was obtained in the CIELab system (L*: lightness; a*: redness and b*: yellowness). Texture profile analysis (TPA) was determined in cooked samples obtained from the cooking loss test. Three meat pieces (1 cm x 1 cm x 2.5 cm) from each sample were evaluated using a texture analyzer (TA.XTplus, Stable Micro Systems, Vienna Court, UK) and the computer software (Texture Exponent 32 (version 1.0.0.68, Stable Micro Systems, Vienna Court, UK). Samples were compressed to 60% (using a cylindrical probe with a flat surface area of 19.85 cm^2^) in a double compression cycle test at a speed of 3.33 mm/s. The parameters determined were hardness (N), springiness (mm), cohesiveness, gumminess (N), and chewiness (N.mm).

### 2.9. Sensory Analysis

The evaluation of sensory properties of patties was carried out in the *Centro Tecnológico de la Carne*, equipped with individual standardized cabins [23], by 66 consumers (29–45 years old from both genders). Unfortunately, the preventive measures against COVID-19 (SARS-CoV-2) limited the presence of a greater number of participants. The following sensory attributes were evaluated: visual aspect (raw patties), odor, texture, taste, and overall acceptability using a 7-point hedonic scale, where a score of 7 represented “highly acceptable” and 1 represented “highly not acceptable”, a score of 4 being the acceptability limit [24]. With the exception of the visual aspect, which was scored in the raw patties, the other sensory attributes were determined in cooked samples. Patties were prepared in a convection oven at 180 °C (Rational Combi Master^®^ Plus CMP61, Landsberg am Lech, Germany) until an internal temperature of 70 °C. Once cooked, the patties were cut into pieces (4 × 4 × 2.5 cm), individually wrapped in foil (coded by three-digit numbers), and served to consumers in disposable plastic dishes [25]. Bread without salt and water was available for panelists to clean their palate between the samples. Moreover, a preference test was performed using a 6-point scale, where 1 corresponded to the least preferred product and 6 to the most preferred [26].

### 2.10. Statistical Analysis

Statistical analyses of the results obtained were performed using the Statgraphics Centurion XVI version 16.1.03 (32-bits) (StatPoint Technologies, Inc., Warrenton, VA, USA). Normal distribution and homogeneity of variance were previously tested (Shapiro–Wilk). The data were evaluated with two-factor analysis of variance (ANOVA) (treatment and storage time as a fixed effect, and replicate as a random effect), or a one-way ANOVA (for chemical composition). Tukey’s test was used to compare the mean values when the ANOVA was significant (*p* < 0.05). Regarding the sensory analysis, consumers were considered as a random effect (each panelist tasted three samples, one for each treatment, in a single session). The statistical evaluation for the preference test was performed using the Friedmann test with Newell and McFarlane tables (α = 0.05). The least significant difference (LSD) test was used to compare the mean values of treatments in the case of significant differences (*p* < 0.05).

## 3. Results and Discussion

### 3.1. Chemical Compositions Results

The results for chemical composition and sodium content of all treatments, as well as the composition of Pd and Hs powders, are shown in Table 1. Pd powder exhibited an interesting protein content in the range reported by other works [6,10], and dietary fiber contents similar to those reported by Bach et al. [27]. Despite there being several methods (oven drying, microwave, or vacuum oven) to obtain a dried ingredient from *Pleurotus djamor*, conventional oven drying yields better nutritional composition with the least degradation, also being economic and easily carried out according to Siti-Nuramira et al. [10]. In the case of Hs, dietary fiber was the main component, reaching a value of 70.2 ± 6.5%, which was similar to that reported by Amaya-Cruz et al. [13]. The increase in total fiber content in Hs by-products compared to roselle powder was attributed by these authors to the leak of simple carbohydrates during the decoction process.

Regarding the chemical composition of patties, the inclusion of Pd and Hs did not significantly affect fat and protein content (*p* > 0.05). The replacement of meat by Pd and Hs hardly modified the protein content even though the protein concentration of Pd was higher than 25%. Bach et al. [27] indicates that *Pleurotus djamor* is an interesting source of essential amino acids (leucine is the limiting amino acid), a complement to vegetables, and an relevant potential alternative to meat in terms of fulfilling the daily essential amino acid requirements. However, the addition of Pd and Hs resulted in significantly (*p* < 0.05) lower moisture values of patties, especially in samples with the highest inclusion of alternative ingredients (Pd7.5-Hs2.5), where the moisture content was reduced to 70.53% compared to 76.12% of the control samples. In general, when mushroom powders have been used in meat products as fat replacers, an increase in moisture has been observed, since the reduction of fat has been usually compensated for with a mixture of mushroom and water [5]. In this case, the substitution of fresh meat (with a high moisture content) by a dry ingredient significantly decreased moisture, which could affect the patty texture. Similarly, ash and fiber contents were significantly affected (*p* < 0.05) by the addition of Pd and Hs, as the non-meat ingredient concentration increased. This effect could be attributed to the higher fiber content naturally found in vegetables and fungi. Fiber from edible mushrooms includes cellulose, chitin, α and β-glucans, and other hemicelluloses such as mannans, xylans, and galactans [28]. Fiber from *Hibiscus* includes carbohydrate polymers such as polysaccharides (lignin and other non-carbohydrate components such as polyphenols, for instance), which could also confer the functional properties of supplemented foods, such as improved regulation of body weight and glucose and lipid metabolism [13].

Regarding sodium content, patties manufactured with Pd and Hs powders also had 0.7% NaCl added to accomplish the low-sodium claim, since burgers or patties usually contain 2–3% NaCl [29]. Consequently, all treatments had low Na content, ranging from 283.81–334.98 mg/100 g. The umami taste of mushrooms related to the presence of glutamic and aspartic amino acids and 5’-nucleotides has made mushrooms a suitable natural flavor enhancer to replace NaCl [30].

### 3.2. Antioxidant Properties of Pd and Hs Ingredients and Patty Lipid Stability

The antioxidant properties and polyphenol content of Pd and Hs methanolic extracts are reported in Table 2. *Hibiscus* calyx has been reported to be rich in bioactive compounds, such as polyphenols, that can exert antioxidant activity [31,32]. In this regard, Hs by-products displayed higher antioxidant properties than oyster mushroom powder. However, the DPPH values (45.83 vs. 98.29 µmol Trolox/g) and phenolic contents (7.00 vs. 15.02 mg GAE/g) obtained in Hs by-products were around half of those reported by Villasante et al. [32] in the original roselle plant. These values were also lower than those reported by Borras-Linares et al. [33] in the extract obtained from the dried calyxes of several varieties of Mexican roselle (24–100 mg GAE/g, 27.4–112 µmol Trolox/g).

According to Amaya-Cruz et al. [31], the polyphenol content may be retained in roselle by-products after the solid–liquid extraction. These authors argued that non-extractable polyphenols with high antioxidant activity remain bound to organic matter (e.g., proteins and polysaccharides) composing this by-product, and are poorly extracted using a decoction extraction method. Moreover, Amaya-Cruz et al. [31] reported polyphenol content (6.83 mg GAE/g) close that to obtained in our study.

Regarding Pd methanolic extract, our result for polyphenol quantification (12.42 mg GAE/g) was higher than the values obtained by Acharya et al. [34] (7.85 mg GAE/g), Jegadeesh et al. [35] (2.25 mg GAE/g) and Nayak et al. [36] (1.14 mg GAE/g) in different varieties of *Pleurotus djamor*.

In light of the antioxidant results found for Pd and Hs ingredients, Hs demonstrated a better antioxidant effect to delay lipid oxidation in beef patties (Figure 2). As expected, TBARs showed a gradual increase over the storage time because of oxidation processes. The less pronounced evolution was observed in Hs5 samples with TBARs in the range of 0.191–0.464 mg MDA/kg and control samples (0.335–0.601 mg MDA/kg). It is relevant to note that these values were below 0.6 mg MDA/kg, considered as the lower limit for the detection of flavor deterioration related to rancidity in meat products [37]. Similar results were also obtained in Hs2 up to day 7 of storage, showing significantly (*p* < 0.05) lower TBARs values than those obtained in the control sample at day 7. Despite the antioxidant properties of roselle by-products attributed to the presence of non-extractable polyphenols such as hydroxycinnamic and hydroxybenzoic acids, flavonols, flavanols, and flavan-3-ols polymers [31], the effect was quite limited in this study.

Despite the reported antioxidant properties attributed to *Pleurotus djamor* [38,39,40] (generally attributed to phenolic compounds), our study indicated that samples containing Pd powder presented significantly (*p* < 0.05) higher or similar TBARs values than control samples (2.42 vs. 0.60 mg MDA/kg at the end of storage, respectively) (Figure 1). The effect of Pd on TBARs values was the opposite of that found by Banerjee et al. [41] in goat nuggets manufactured with enoki mushroom stem waste powder. These authors obtained an aqueous extract that had lower polyphenolic content than the Pd powder used in the present study (6.3 vs. 12.42 mg GAE/g, respectively). This difference may be explained by the potential pro-oxidant effect of Pd in the beef patties, since there was a significant (*p* > 0.05) increase in TBARs compared to control samples. This trend increased throughout storage, almost reaching the upper limit value (2.5 mg MDA/kg) for the sensory perception of rancidity at the end of the storage time [42]. This absence of an antioxidant effect in mushroom powder was also observed in our previous work [7], which was attributed to the drying conditions of the mushrooms (60 °C). In the present study, drying temperature was reduced (40 °C) but did not prevent the loss of antioxidant activity of Pd powder.

Its combination with the lowest dose of Hs (Pd5-Hs2) allowed us to improve the antioxidant effect of Pd. However, it was not enough to reach the oxidative stability achieved in the patties treated with Hs2 (1.79 vs. 1.02 mg MDA/kg at day 12, respectively). The TBARS values of Pd5-Hs2 patties were also higher than that obtained from control samples. Moreover, the increase in the amount of Pd favored the lipid oxidation in patties. The combination of Pd with Hs (Pd7.5-Hs2.5) led to unacceptable values at 7 days of storage (>2.5 mg MDA/kg).

### 3.3. pH and Microbiological Results

The effect of Pd and Hs ingredients on the evolution of pH and microbial counts of beef patties during storage is shown in Table 3. The pH was significantly (*p* < 0.05) affected by the roselle by-product. In the case of Pd, a similar pH to control sample (around 6.0) was obtained. Previous results have shown that the addition of 2% of roselle powder in beef patties resulted in a pH between 5.20–5.42 [32], while water roselle extracts (pH 2.3) added at 1% reduced pH to 4.96 [43]. In this case, despite the previous decoction, several non-extractable polyphenols (associated with protein and fiber contents, as well as some remaining acids) reduced pH, especially in samples with 5% of Hs where the pH dropped significantly below 5.0.

Microbiological analysis indicated that no significant differences were observed between the formulations for the different microbial groups at the beginning of the storage, except for Pd5 in molds and yeast counts (Figure 2). TVC ranged from 3.69 to 4.38 Log CFU/g, which are usual considering the intense manipulation during the elaboration process [32]. In general, all the microbial counts increased during storage, except in the Hs5 treatment. After 4 days of storage, TVC and *Pseudomonas* exceeded 8 Log CFU/g in control samples. This result indicates that control samples exceeded the legal limits (10^6^ CFU/g) [44] and became unsuitable for consumption. In contrast, lower counts were observed for the other formulations (5.30–6.77 Log CFU/g and 5.65–6.65 Log CFU/g for TVC and *Pseudomonas*, respectively).

The antimicrobial activity of roselle against several foodborne pathogenic bacteria (Gram-positive and Gram-negative) has been well documented [14,32,33,45,46], but the information available on the antimicrobial effect against spoilage micro-organisms is limited. In our study, the antimicrobial effect of Hs was clearly shown after day 4 in TVC and *Pseudomonads* counts (Table 3). In the case of enterobacteria counts, differences among treatments were observed after day 7. Among all treatments, Hs5 (lowest pH and high Hs by-product concentration) patties exhibited the lowest *Pseudomonas*, TVC enterobacteria, and mold and yeast counts, especially after 7 days of storage. 

Samples with both Hs and Pd (Pd5-Hs2 and Pd7.5-Hs2.5) showed intermediate values between control and Hs5 samples counts. Finally, LAB counts were hardly affected by Hs powder, although some differences were observed among treatments. Control samples and Hs5 samples presented the lowest LAB counts (over 5 Log CFU/g) at the end of storage, while the other formulations were around 6 Log CFU/g. Additionally, the intense microbial activity to promote their growth may explain the increase of pH due to the degradation of meat proteins by endogenous and spoilage bacteria enzymes [32,47].

### 3.4. Cooking Loss

The Pd and Hs powders, as well as the related modification of pH, also affected the cooking losses (Figure 3). According to the results shown in Table 1, the moisture reduction in patties due to the replacement of meat by Ps and Hs powders did not correspond with a diminution of cooking losses. In fact, only samples with 5% (Pd5) and 7.5% (Pd7.5-Hs2.5) of Pd presented significantly lower cooking losses (*p* < 0.05) than control samples (9.46% and 16.64% vs. 22.79% at the beginning of storage, respectively). The addition of 5% of other edible mushrooms (*Agaricus bisporus* and *Pleurotus ostreatus*) also contributed to the reduction of cooking losses in beef patties [7].

On the contrary, the inclusion of *Hibiscus* by-product alone seems to have limited capacity to retain meat components during cooking. Hs2 patties had lower capacity to prevent the loss of components during cooking in comparison to control samples (28.76%). In the case of Hs5, a similar cooking loss to control patties was obtained. Although it has been reported that the presence of dietary fiber from different vegetable sources in meat products can help to increase the water-holding capacity and improve the technological properties of the meat products [48], it seems that *Hibiscus* by-products would not be an adequate natural ingredient for that purpose, despite their high dietary fiber content.

The several acids naturally found in roselle calyxes, such as malic, citric, hydroxycitric, tartaric, and hibiscus acids, as well as some phenolic acids, such as caffeic, protocatechuic, chlorogenic, or hibiscus acids, give a low pH (around 2 units) to roselle aqueous extracts [46,49,50]. Therefore, despite the extraction with water of several biocompounds, including some acids, the remaining presence of acids is enough to provoke a pH diminution close to the isoelectric point of meat proteins, especially in the Hs2 patties (pH = 5.23), reducing the water holding capacity and increasing the cooking losses. In our experiment, Hs2 samples presented values near 30% during the whole period. This increase in cooking losses by the addition of roselle extracts in beef patties has also been reported by Perez-Baez et al. [24].

In the case of samples containing both Hs and Pd powders, a reduction of pH near to the isoelectric point of meat proteins was reached, but the presence of Pd components (with a higher capacity to retain meat water and minor components) compensated the reduced capacity of myofibrilar proteins. Consequently, cooking loss of Pd5-Hs2 and Pd7.5-Hs2.5 was improved in relation to Hs2 patties. It is relevant to comment that cooking losses significantly increased in Pd5-Hs2 and Pd7.5-Hs2.5 treatments only at day 12 (values over 20%). These values are close to the other samples. These results strengthen the hypothesis that Pd powder can compensate for the cooking loss associated with the addition of Hs, especially the effect observed on Hs2 patties.

### 3.5. Color and Texture

Almost all treatments containing Pd and/or Hs reduced the L*, a*, and b* values of patties in relation to control at day 0 (Table 4). The exception was the Pd5 treatment, which had initially significantly (*p* < 0.05) higher a* and b* values than those of the control samples (20.12 and 17.92 vs. 17.51 and 15.04, respectively). Banerjee et al. [41] also observed a slight increase in L* and a* values due to replacement of meat by 6% of *Flammulina velutipes* in goat meat nuggets.

In a previous study, the partial replacement of fat and salt by *Agaricus bisporus* flour significantly affected the color of beef patties [7]. However, in this study our results indicated that Pd powder seems to have potential to avoid great color modifications of original sample patties, unlike Hs powder. Hs by-product led to significantly lower color parameters (*p* < 0.05) even when Pd powder was present (darker colorations). The lowest L*, a*, and b* parameters at day 0 were observed in Hs5 samples (31.88 vs. 41.69, 7 vs. 17.51 and 5.40 vs. 15.04 in Hs5 and control samples, respectively). In other words, Hs5 samples were darker and less red and yellow than control (Figure 1). The effect of Hs powder in color may be explained by the high content of anthocyanins of *Hibiscus* (mainly delphinidin-3-O-sambubioside, cyanidin-3-O-sambubioside, and delphinidin-3-glucoside responsible for the bright red color) that have remained in the Hs by-product and darkened the meat product [24].

During the storage, the main changes were observed in the a* parameter for control, Pd5, and Hs2 samples, which significantly decayed (above 35%). This effect was probably due to the natural oxidation of myoglobin during storage, specifically the iron oxidation from the heme group [51]. On the contrary, L* and b* values remained quite stable or with slight modifications until the end of storage. According to the results, it seems that Hs by-product components (such as anthocyanins) ameliorated the discoloration process during cold storage.

The addition of Pd and Hs significantly (*p* < 0.05) affected the textural properties of patties (Table 5). This effect is in agreement with the results found in other studies with other non-meat ingredients added to meat products [24,52]. The hardness of patties with the mixtures Pd-Hs or with 2% of Hs was not significantly different from control samples. The opposite effect was observed by Kurt and Gençcelep [52], who observed an increase in the hardness of meat emulsions with the addition of *Agaricus bisporus* powder in doses of up to 2% (72.70 vs. 56.86 g for cooked meat emulsions included with 2% mushroom powder and control samples, respectively).

However, samples with 5% Pd (Pd5) and 5% of Hs (Hs5) exhibited the lowest values of hardness (64.00 N and 57.37 N, respectively). The same trend was noted by the aforementioned authors, who found that the addition of 3% mushroom powder decreased textural properties [52]. In another experiment in Frankfurt-type sausages, doses of roselle extract higher than 6% were necessary to observe the same effect [24]. This behavior was also observed in respect of gumminess and chewiness. In the case of Pd5 samples, the softer texture could be due to the lower cooking loss and the ability of *Pleurotus* proteins and fibers to retain water [5,53]. It is worth remembering that Hs5 samples presented the lowest pH (Figure 3a) that would affect the formation and stability of protein gels within the meat matrix, limiting the gelling capacity of these proteins and reducing hardness [24]. This means that the addition or substitution of ingredients by vegetables is commonly associated with softer textures [41].

Regarding springiness and cohesiveness, Hs samples did not significantly differ from control samples (0.765 and 0.716 mm vs. 0.778 mm, and 0.576 and 0.552 vs. 0.595 for springiness and cohesiveness values of Hs2, Hs5 and control samples, respectively). On the contrary, the inclusion of Pd or its combinations with Hs resulted in significant reductions of these parameters. The significant reduction in springiness and cohesiveness due to the incorporation of edible mushroom powders was also reported previously [7], and it is also a common effect when non-meat ingredients are added to meat products, resulting in less cohesive structures [47].

### 3.6. Sensorial Analysis

The sensory scores of the reformulated patties are presented in Table 6. The incorporation of Hs residue (rich in anthocyanins) negatively affected the visual aspect of the patties. It is known that the stability of natural pigments in foods is conditioned by several factors, such as pH, temperature, oxygen concentration, UV radiation, and water activity [54]. In the case of roselle extracts, the stability of its anthocyanins (responsible for the characteristic color of roselle) has been associated with changes in pH and the interaction with matrix components [54]. Similar to observed in our study, the modification of color caused by Hs powder or its extracts has been previously observed in other meat products, but it is not always appreciable [55,56]. It is worth considering that the thermal treatment during the decoction process could have favored the degradation of the remaining pigments, converting them into colorless chalcones, and would also lead to the darkening of other compounds [57]. This would explain the lowest scores attributed to samples with 5% of Hs (2.1 vs. 5.9 for Hs5 and control samples, respectively) in the visual aspect (*p* < 0.05).

On the contrary, the pink oyster hardly modified sensorial properties, presenting similar values to control samples (*p* > 0.05). In this regard, the addition of 5% of Pd resulted in a not significantly different visual aspect from control samples (6.1 vs. 5.9, respectively). Moreover, Pd5 samples also presented the highest sensorial values for odor (5.1 vs. 4.1 for Pd5 and Hs5, respectively), texture (5.6 vs. 3.0 for Pd5 and Pd7.5-Hs2.5, respectively), taste (5.5 vs. 2.8 for Pd5 and Pd7.5-Hs2.5, respectively), and overall acceptance (5.1 vs. 2.5 for Pd5 and Pd7.5-Hs2.5, respectively), which were also similar to those attributed to control samples. It is important to note that the sensory acceptance of the product is related not only to the mushroom species chosen and the concentration added, but also to the familiarization of the consumer with the taste [7,58]. The incorporation of *Flammulina velutipes*, *Lentinula edodes*, and *Volvariella volvacea* powders in some meat products has also been sensorially accepted and even received higher scores for overall acceptance in relation to their control treatments [41,59,60,61].

Hs2 and Pd5-Hs2 presented values close to the acceptability limit (scores higher than 4), while samples with the highest concentration of *Hibiscus* by-product (Hs5 and Pd7.5-Hs2.5) received significantly (*p* < 0.05) lower scores in texture, taste, and overall appearance. The results obtained for these treatments may be explained by the reduction in pH, inducing the denaturation of myofibrillar proteins and decreasing water retention, thus affecting the texture and juiciness of the patties [43].

The inclusion of Pd seems to revert, to some extent, the decrease in water retention induced by Hs powder. However, the patties had a pasty texture (especially on Pd7.5-Hs2.5 treatment), which was not well accepted by the panelists (3.0 vs. 5.7 for Pd7.5-Hs2.5 and control samples, respectively). Friedman´s test indicated consistent differences among treatments (F test) > F (α = 0.05). According to the total scores of preferences (number in brackets in Table 7), the results of the LSD test showed two groups: one group, the most preferred, formed by the control, Pd5 and Hs2 patties; and the other group (the least preferred) comprising the other treatments. These results are in agreement with those obtained in the acceptance test (Table 6).

## 4. Conclusions

Despite the high antioxidant and antimicrobial properties of roselle by-products in beef patties, a decrease in texture and sensorial characteristics at high concentrations was unavoidable, mainly because of the pH decrease and color modification. On the contrary, pink oyster mushroom powder offered exceptional sensory properties, similar to control samples, but with limited antimicrobial effect and no antioxidant effect on lipid stability. Although the combination of oyster mushrooms with Hs by-products could revert some of the negative aspects of roselle by-products, the formulation should be optimized to avoid consumer rejection. Therefore, Hs powder (to a maximum of 2%) could be used in the production of low-sodium beef patties.

## Figures and Tables

**Figure 1 foods-12-00391-f001:**
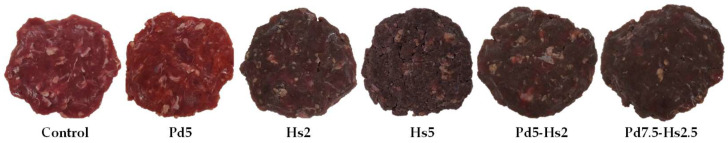
Visual aspect of the beef burgers formulated after the preparation and considering the six treatments: control, Pd5 (5% *Pleurotus djamor* powder), Hs2 (2% *Hibiscus sabdariffa* powder), Hs5 (5% *Hibiscus sabdariffa* powder), Pd5-Hs2 (5% *Pleurotus djamor* powder with 2% *Hibiscus sabdariffa* powder), and Pd7.5-Hs2.5 (7.5% *Hibiscus sabdariffa* powder with 2.5% *Hibiscus sabdariffa* powder).

**Figure 2 foods-12-00391-f002:**
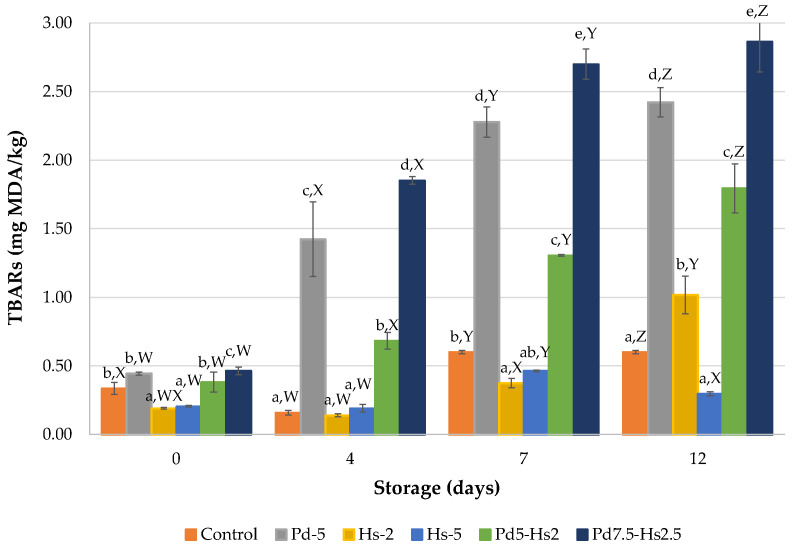
TBARs results (mg MDA/kg) of formulations with Pd and Hs powders. ^a–e^ Mean values for each day with different letters differ significantly (*p* < 0.05). ^W–Z^ Mean values for each formulation with different letters differ significantly (*p* < 0.05).

**Figure 3 foods-12-00391-f003:**
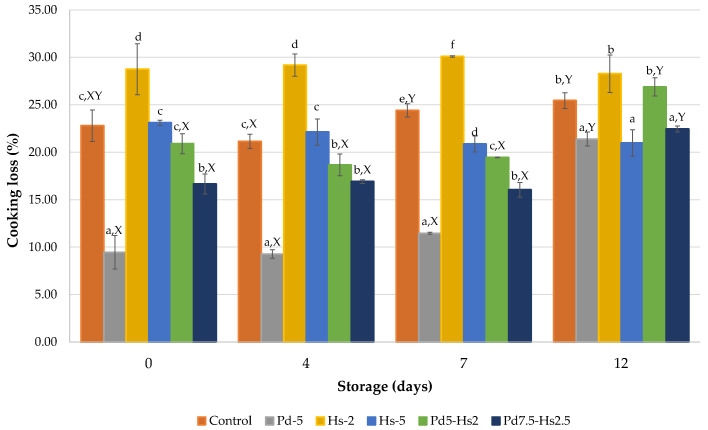
Cooking losses (%) of formulations with Pd and Hs powders. ^a–f^ Mean values for each day with different letters differ significantly (*p* < 0.05). ^X,Y^ Mean values for each formulation with different letters differ significantly (*p* < 0.05).

**Table 1 foods-12-00391-t001:** Proximate composition (%) and Na content (mg/100 g) of Pd and Hs ingredients, and formulated beef patties.

	Moisture	Fat	Protein	Ash	Dietary Fiber	Na
Pd	6.0 ± 0.2	0.69 ± 0.02	25.6 ± 0.2	5.8 ± 0.4	28 ± 8	69 ± 13
Hs	5.18 ± 0.08	1.23 ± 0.01	8.4 ± 0.3	4.3 ± 0.1	70 ± 7	145 ± 4
Control	76.1 ± 0.4 ^d^	3.6 ± 0.3	18.1 ± 0.1	1.49 ± 0.03 ^a^	ND	311 ± 3 ^a,b^
Pd5	72.6 ± 0.2 ^b^	3.0 ± 0.2	18.8 ± 0.7	1.5 ± 0.1 ^a^	1.2 ± 0.3 ^a^	335 ± 30 ^b^
Hs2	74.9 ± 0.7 ^c^	4.0 ± 0.4	18.2 ± 0.2	1.52 ± 0.01 ^a^	1.3 ± 0.2 ^a^	335 ± 10 ^b^
Hs5	72.9 ± 0.6 ^b^	3.9 ± 0.3	17.8 ± 0.3	1.84 ± 0.08 ^b,c^	3.2 ± 0.4 ^b,c^	284 ± 5 ^a^
Pd5-Hs2	72.3 ± 0.6 ^b^	3.4 ± 0.7	18.41 ± 0.01	1.78 ± 0.01 ^b^	2.5 ± 0.4 ^b^	302 ± 2 ^a^
Pd7.5-Hs2.5	70.53 ± 0.08 ^a^	3.6 ± 0.1	18.8 ± 0.1	1.95 ± 0.01 ^c^	3.5 ± 0.6 ^c^	299 ± 1 ^a^

Results are expressed as mean value ± standard deviation. ND: not detected. ^a–c^ Different letters in each column indicate significant differences (*p* < 0.05).

**Table 2 foods-12-00391-t002:** Antioxidant properties and polyphenol content of Pd and Hs methanolic extracts.

	DPPH (mg Trolox/g)	ABTS (mg Trolox/g)	FRAP (mmol FeSO_4_/100 g)	Polyphenol Content (mg GAE/g)
Pd	8.13 ± 0.05	8.60 ± 0.03	2.18 ± 0.04	12.4 ± 0.2
Hs	11.5 ± 0.3	21.8 ± 0.5	3.88 ± 0.08	7.00 ± 0.06

Results are expressed as mean value ± standard deviation.

**Table 3 foods-12-00391-t003:** Evolution of pH and microbial counts (Log CFU/g) of cold stored beef patties.

		Storage Time (Days)			
	Treatment	1	4	7	12
**pH**	C	5.97 ± 0.05 ^d,W^	6.29 ± 0.01 ^d,X^	5.99 ± 0.06 ^e,W^	6.20 ± 0.01 ^e,X^
	Pd5	6.01 ± 0.02 ^d,Y^	6.25 ± 0.01 ^d,Z^	5.88 ± 0.01 ^d,X^	5.72 ± 0.01 ^d,W^
	Hs2	5.23 ± 0.01 ^b,W^	5.48 ± 0.01 ^b,X^	5.29 ± 0.01 ^b,W^	5.56 ± 0.06 ^c,X^
	Hs5	4.65 ± 0.02 ^a,W^	4.87 ± 0.01 ^a,X^	4.74 ± 0.01 ^a,W^	4.97 ± 0.07 ^a,X^
	Pd5-Hs2	5.37 ± 0.01 ^c,W^	5.74 ± 0.08 ^c,Y^	5.59 ± 0.04 ^c,X,Y^	5.52 ± 0.08 ^c,WX^
	Pd7.5-Hs2.5	5.35 ± 0.01 ^c,X^	5.67 ± 0.03 ^c,Y^	5.58 ± 0.01 ^c,Y^	5.15 ± 0.06 ^b,W^
**TVC**	C	4.1 ± 0.4 ^W^	8.3 ± 0.2 ^e,X^	8.11 ± 0.05 ^e,X^	8.31 ± 0.05 ^e,X^
	Pd5	4.1 ± 0.4 ^W^	6.8 ± 0.1 ^d,X^	7.40 ± 0.02 ^d,X^	7.2 ± 0.2 ^cd,X^
	Hs2	3.85 ± 0.09 ^W^	6.15 ± 0.04 ^bc,X^	7.02 ± 0.09 ^c,Y^	7.5 ± 0.3 ^d,Z^
	Hs5	3.70 ± 0.07 ^W^	5.3 ± 0.4 ^a,Y^	4.4 ± 0.1 ^a,X^	4.69 ± 0.01 ^a,X^
	Pd5-Hs2	4.0 ± 0.2 ^W^	6.3 ± 0.1 ^c,X^	6.9 ± 0.2 ^c,Y^	6.96 ± 0.06 ^c,Y^
	Pd7.5-Hs2.5	4.4 ± 0.2 ^W^	5.81 ± 0.03 ^b,X^	6.00 ± 0.01 ^b,X^	6.00 ± 0.01 ^b,X^
** *Pseudomonas* **	C	5.02 ± 0.02 ^W^	8.59 ± 0.07 ^d,Z^	7.3 ± 0.3 ^e,X^	8.20 ± 0.02 ^d,Y^
	Pd5	4.93 ± 0.06 ^W^	6.65 ± 0.08 ^c,X^	6.69 ± 0.01 ^d,XY^	7.1 ± 0.3 ^c,Y^
	Hs2	4.95 ± 0.01 ^W^	6.4 ± 0.2 ^bc,Y^	6.05 ± 0.07 ^c,X^	6.2 ± 0.1 ^b,X,Y^
	Hs5	4.8 ± 0.3 ^X^	5.7 ± 0.5 ^a,Y^	3.96 ± 0.01 ^a,W^	3.89 ± 0.04 ^a,W^
	Pd5-Hs2	5.1 ± 0.2 ^W^	6.4 ± 0.2 ^b,c,X^	5.96 ± 0.02 ^b,c,X^	6.3 ± 0.2 ^b,X^
	Pd7.5-Hs2.5	5.0 ± 0.3 ^W^	5.97 ± 0.01 ^a,b,XY^	5.7 ± 0.1 ^b,X^	6.20 ± 0.09 ^b,Y^
**LAB**	C	4.0 ± 0.1 ^W^	5.4 ± 0.1 ^d,Y^	4.9 ± 0.1 ^a,b,X^	5.2 ± 0.1 ^a,XY^
	Pd5	3.98 ± 0.02 ^W^	4.60 ± 0.02 ^c,X^	5.49 ± 0.05 ^c,Y^	6.0 ± 0.3 ^b,Y^
	Hs2	3.99 ± 0.02 ^W^	4.5 ± 0.2 ^bc,X^	5.1 ± 0.2 ^b,Y^	6.2 ± 0.1 ^b,Z^
	Hs5	3.88 ± 0.01 ^W^	4.2 ± 0.1 ^a,X^	4.71 ± 0.07 ^a,Y^	5.22 ± 0.06 ^a,Z^
	Pd5-Hs2	4.0 ± 0.1 ^W^	4.67 ± 0.04 ^c,X^	4.9 ± 0.1 ^ab,X^	6.2 ± 0.4 ^b,Y^
	Pd7.5-Hs2.5	4.03 ± 0.02 ^W^	4.37 ± 0.01 ^ab,X^	5.04 ± 0.06 ^b,Y^	5.95 ± 0.07 ^b,Z^
**Enterobacteria**	C	3.5 ± 0.1 ^W^	4.0 ± 0.7 ^W^	5.8 ± 0.2 ^e,X^	7.02 ± 0.08 ^e,Y^
	Pd5	3.3 ± 0.3	4 ± 1	5.23 ± 0.01 ^d^	5.8 ± 0.6 ^d^
	Hs2	3.0 ± 0.3 ^W^	4.4 ± 0.1 ^Y^	3.8 ± 0.1 ^b,X^	4.00 ± 0.01 ^b,XY^
	Hs5	3.04 ± 0.06 ^X^	3.70 ± 0.06 ^Y^	3.00 ± 0.01 ^a,X^	1.2 ± 0.2 ^a,W^
	Pd5-Hs2	3.06 ± 0.03	4 ± 1	4.36 ± 0.08 ^c^	5.0 ± 0.3 ^c^
	Pd7.5-Hs2.5	3.4 ± 0.4	4.1 ± 0.4	4.1 ± 0.4 ^bc^	4.9 ± 0.2 ^c^
**Molds and yeasts**	C	4.8 ± 0.1 ^b^	4.9 ± 0.9	5.32 ± 0.01 ^b^	6.0 ± 0.2 ^c^
	Pd5	3.7 ± 0.5 ^a,W^	5.0 ± 0.3 ^X^	5.28 ± 0.06 ^b,X^	5.6 ± 0.2 ^c,X^
	Hs2	4.6 ± 0.2 ^b^	5.1 ± 0.1	4.7 ± 0.4 ^b^	4.8 ± 0.4 ^b^
	Hs5	4.47 ± 0.06 ^b,X^	5.75 ± 0.04 ^Y^	3.3 ± 0.4 ^a,W^	3.3 ± 0.4 ^a,W^
	Pd5-Hs2	4.46 ± 0.04 ^b^	4.9 ± 0.4	5.2 ± 0.2 ^b^	4.8 ± 0.4 ^b^
	Pd7.5-Hs2.5	4.6 ± 0.2 ^b^	4.6 ± 0.3	5.0 ± 0.4 ^b^	5.3 ± 0.1 ^bc^

Results are expressed as mean value ± standard deviation. ^a–e^ Means in the same column with a different letter are significantly different (*p* < 0.05). ^W–Z^ Means in the same row with a different letter are significantly different (*p* < 0.05).

**Table 4 foods-12-00391-t004:** CIELab color parameters from different beef patty formulations.

		Storage (Days)			
	Treatment	0	4	7	12
L*	Control	41.7 ± 0.4 ^b^	43 ± 3 ^c^	43.3 ± 0.3 ^c^	42.4 ± 0.5 ^c^
	Pd5	42.3 ± 0.5 ^b,Y^	40.9 ± 0.3 ^b,c,X^	42.5 ± 0.6 ^c,Y^	43.4 ± 0.1 ^c,Y^
	Hs2	39 ± 2 ^b^	37 ± 2 ^a,b^	38 ± 3 ^b^	39 ± 2 ^b^
	Hs5	32 ± 1 ^a^	34.6 ± 0.6 ^a^	35 ± 1 ^a,b^	34.3 ± 0.2 ^a^
	Pd5-Hs2	34 ± 3 ^a^	34.2 ± 0.8 ^a^	35.2 ± 0.6 ^a,b^	36.8 ± 0.9 ^b^
	Pd7.5-Hs2.5	33 ± 1 ^a,X^	37 ± 2 ^a,b,Y^	33.5 ± 0.3 ^a,X^	37 ± 1 ^b,Y^
a*	Control	17.5 ± 0.3 ^c,Y,Z^	15 ± 2 ^c,Y^	19.0 ± 0.3 ^d,Z^	11.3 ± 0.9 ^d,X^
	Pd5	20 ± 1 ^d,Z^	17 ± 1 ^c,Y^	12.5 ± 0.6 ^c,X^	11.03 ± 0.04 ^d,X^
	Hs2	9.1 ± 0.9 ^b,Y^	6.89 ± 0.08 ^a,X^	6.7 ± 0.4 ^a,X^	5.50 ± 0.01 ^a,X^
	Hs5	7.0 ± 0.3 ^a^	6.8 ± 0.3 ^a^	6.3 ± 0.7 ^a^	5.3 ± 0.3 ^a^
	Pd5-Hs2	8.5 ± 0.3 ^a,b^	8.7 ± 0.6 ^ab^	8.4 ± 0.2 ^b^	7.88 ± 0.08 ^b^
	Pd7.5-Hs2.5	9.0 ± 0.3 ^a^	9.5 ± 0.8 ^b^	8.4 ± 0.4 ^b^	9.2 ± 0.5 ^c^
b*	Control	15.0 ± 0.8 ^c^	16 ± 2 ^c^	17.7 ± 0.6 ^d^	13.4 ± 0.7 ^c^
	Pd5	18 ± 1 ^d^	17.2 ± 0.5 ^c^	19 ± 2 ^d^	19.7 ± 0.5 ^f^
	Hs2	10.3 ± 0.8 ^b^	10 ± 1 ^b^	9.7 ± 0.9 ^b^	12.3 ± 0.2 ^b^
	Hs5	5.4 ± 0.3 ^a^	5.6 ± 0.7 ^a^	7 ± 2 ^a^	5.9 ± 0.5 ^a^
	Pd5-Hs2	9 ± 1 ^b,X^	13 ± 1 ^b,Y^	14.5 ± 1.1 ^c,Y^	14.9 ± 0.3 ^d,Y^
	Pd7.5-Hs2.5	11 ± 2 ^b,X^	18 ± 1 ^c,Z^	14.3 ± 0.4 ^c,Y^	16.8 ± 0.3 ^e,Y^

Results are expressed as mean value ± standard deviation. ^a–f^ Means in the same column with a different letter are significantly different (*p* < 0.05). ^X–Z^ Means in the same row with a different letter are significantly different (*p* < 0.05).

**Table 5 foods-12-00391-t005:** Texture parameters of beef patties.

		Storage (Days)			
	Treatment	0	4	7	12
**Hardness (N)**	Control	85 ± 4 ^b^	95 ± 16 ^b,c^	97 ± 19 ^b^	86 ± 4 ^c^
	Pd5	64 ± 2 ^a^	75 ± 4 ^a,b^	62 ± 11 ^a^	65 ± 2 ^a,b^
	Hs2	94 ± 6 ^b^	102 ± 2 ^c^	100 ± 5 ^b^	92 ± 4 ^c^
	Hs5	57 ± 11 ^a^	67 ± 13 ^a^	59 ± 9 ^a^	60 ± 7 ^ab^
	Pd5-Hs2	90.40 ± 0.03 ^b^	86 ± 10 ^a–c^	69 ± 14 ^a^	68 ± 3 ^b^
	Pd7.5-Hs2.5	88.0 ± 0.8 ^b,Y^	98 ± 8 ^bc,Y^	85 ± 5 ^ab,Y^	57.5 ± 0.2 ^a,X^
**Springiness (mm)**	Control	0.78 ± 0.01 ^c^	0.780 ± 0.001 ^c^	0.79 ± 0.02 ^d^	0.78 ± 0.03 ^d,e^
	Pd5	0.65 ± 0.04 ^a,b^	0.720 ± 0.008 ^b^	0.70 ± 0.04 ^b,c^	0.728 ± 0.002 ^c^
	Hs2	0.765 ± 0.001 ^c^	0.79 ± 0.04 ^c^	0.78 ± 0.04 ^d^	0.82 ± 0.03 ^e^
	Hs5	0.72 ± 0.01 ^c^	0.77 ± 0.02 ^b,c^	0.73 ± 0.01 ^c,d^	0.755 ± 0.003 ^c,d^
	Pd5-Hs2	0.65 ± 0.04 ^b^	0.627 ± 0.001 ^a^	0.647 ± 0.006 ^a,b^	0.66 ± 0.01 ^b^
	Pd7.5-Hs2.5	0.58 ± 0.02 ^a^	0.61 ± 0.03 ^a^	0.62 ± 0.05 ^a^	0.58 ± 0.01 ^a^
**Cohesiveness**	Control	0.60 ± 0.01 ^b,X^	0.602 ± 0.004 ^c,X^	0.631 ± 0.006 ^f,Y^	0.62 ± 0.01 ^d,Y^
	Pd5	0.44 ± 0.05 ^a^	0.48 ± 0.04 ^b^	0.487 ± 0.008 ^c^	0.54 ± 0.01 ^b^
	Hs2	0.58 ± 0.03 ^b,X^	0.589 ± 0.006 ^c,X^	0.600 ± 0.004 ^e,X^	0.64 ± 0.01 ^d,Y^
	Hs5	0.55 ± 0.03 ^b^	0.555 ± 0.003 ^c^	0.57 ± 0.02 ^d^	0.565 ± 0.008 ^c^
	Pd5-Hs2	0.473 ± 0.008 ^a,X^	0.45 ± 0.02 ^a,b,X^	0.46 ± 0.02 ^b,X^	0.528 ± 0.001 ^b,Y^
	Pd7.5-Hs2.5	0.43 ± 0.01 ^a,X^	0.432 ± 0.004 ^a,X^	0.424 ± 0.008 ^a,X^	0.473 ± 0.005 ^a,Y^
**Gumminess (N)**	Control	51 ± 2 ^c,d^	57 ± 9 ^b^	61 ± 13 ^b^	54 ± 1 ^c^
	Pd5	28 ± 4 ^a^	36 ± 1 ^a^	30 ± 7 ^a^	35 ± 2 ^b^
	Hs2	54 ± 6 ^d^	60 ± 2 ^b^	60 ± 4 ^b^	59 ± 1 ^c^
	Hs5	32 ± 8 ^a^	37 ± 7 ^a^	34 ± 7 ^a^	35 ± 5 ^b^
	Pd5-Hs2	42.4 ± 0.4 ^b,c^	38 ± 6 ^a^	32 ± 8 ^a^	36 ± 2 ^b^
	Pd7.5-Hs2.5	37.8 ± 0.7 ^a,b,Y^	42 ± 4 ^a,Y^	36 ± 3 ^a,Y^	27.3 ± 0.6 ^a,X^
**Chewiness (N** **⋅mm)**	Control	39.3 ± 0.5 ^c^	45 ± 7 ^b^	48 ± 9 ^b^	42 ± 2 ^c^
	Pd5	18 ± 4 ^a^	25.9 ± 0.6 ^a^	21 ± 4 ^a^	26 ± 2 ^b^
	Hs2	41 ± 5 ^c^	47 ± 3 ^b^	46.42 ± 0.06 ^b^	48.2 ± 0.5 ^d^
	Hs5	23 ± 6 ^a,b^	28 ± 6 ^a^	25 ± 5 ^a^	27 ± 4 ^b^
	Pd5-Hs2	28 ± 2 ^b^	24 ± 4 ^a^	21 ± 5 ^a^	24 ± 2 ^b^
	Pd7.5-Hs2.5	22 ± 1 ^ab,Y^	26 ± 2 ^a,Y^	23 ± 3 ^a,Y^	16.0 ± 0.6 ^a,X^

Results are expressed as mean value ± standard deviation. ^a–e^ Means in the same column with a different letter are significantly different (*p* < 0.05). ^X–Z^ Means in the same row with a different letter are significantly different (*p* < 0.05).

**Table 6 foods-12-00391-t006:** Acceptance test results of beef patties evaluated at day 0.

Attributes	Treatments						SEM
	Control	Pd5	Hs2	Hs5	Pd5-Hs2	Pd7.5-Hs2.5	
Visual aspect	5.9 ^b^	6.1 ^b^	3.5 ^a^	2.1 ^a^	2.7 ^a^	2.9 ^a^	0.342
Odor	5.4 ^b^	5.1 ^b^	4.6 ^a,b^	4.1 ^a^	4.6 ^a,b^	4.5 ^a,b^	0.235
Texture	5.7 ^c^	5.6 ^b,c^	4.8 ^b,c^	3.2 ^a^	4.0 ^a,b^	3.0 ^a^	0.386
Taste	6.0 ^c^	5.5 ^c^	4.8 ^b,c^	3.3 ^a^	4.0 ^a,b^	2.8 ^a^	0.313
Overall acceptability	5.7 ^c^	5.1 ^c^	4.6 ^b,c^	2.9 ^a^	3.6 ^a,b^	2.5 ^a^	0.343

^a–c^ Mean values in the same row (corresponding to the same parameter) with different letters differ significantly (*p* < 0.05; Duncan test); SEM: standard error of the mean.

**Table 7 foods-12-00391-t007:** Preference test results of beef patties.

	Sample Most Preferred					Sample Least Preferred
Visual aspect	Pd5 (59)	Control (57)	Hs2 (43)			
			Hs2 (43)	Pd5-Hs2 (32)	Pd7.5-Hs2.5 (26)	
					Pd7.5-Hs2.5 (26)	Hs5 (14)
	F_test_ = 41.08 > F(α = 0.05) = 10.79					
Odor	Control (51)	Pd5 (47)	Hs2 (39)	Hs5 (24)	Pd5-Hs2 (39)	Pd7.5-Hs2.5 (31)
	F_test_ = 12.87 > F(α = 0.05) = 10.79					
Texture	Control (55)	Pd5 (54)	Hs2 (43)			
			Hs2 (43)	Pd5-Hs2 (36)		
				Pd5-Hs2 (36)	Hs5 (22)	Pd7.5-Hs2.5 (21)
	F_test_ = 29.03 > F(α = 0.05) = 10.79					
Taste	Control (60)	Pd5 (52)	Hs2 (45)			
			Hs2 (45)	Pd5-Hs2 (33)		
				Pd5-Hs2 (33)	Hs5 (23)	Pd7.5-Hs2.5 (18)
	F_test_ = 35.78 > F(α = 0.05) = 10.79					
Global preference	Control (58)	Pd5 (53)	Hs2 (44)			
		Pd5 (53)	Hs2 (44)	Pd5-Hs2 (37)		
				Pd5-Hs2 (37)	Hs5 (22)	
					Hs5 (22)	Pd7.5-Hs2.5 (17)
	F_test_ = 35.26 > F(α = 0.05) = 10.79					

Samples in the same row (orange) do not have significant differences (*p* > 0.05); samples in different rows have significant differences (*p* < 0.05). Numbers in brackets are Σ score.

## Data Availability

All data are presented in the manuscript.

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
