# Peer review of "Effect of Partial Meat Replacement by Hibiscus sabdariffa By-Product and Pleurotus djamor Powder on the Quality of Beef Patties"

_foods, 2023, doi:10.3390/foods12020391_

Round 1

Reviewer 1 Report

The aim of this study was to investigate the impacts of the replacement of meat by Pleurotus djamor and Hibiscus byproduct powders on the physico-chemical, microbiological, and sensory properties of low-salt beef patties. The experimental design is reasonable, and most of the data are also reliable. This may provide some new insights into the application of low salt beef pie. However, the article has some problems and must be carefully revised.

(1) The author attempts to demonstrate why Pleurotus djamor and Hibiscus byproduct may be used to enhance the quality of meat products. The introduction, however, is inadequate. The author did not explain why both ingredients were added to the meat pie at once. Does synergy exist? What are the combined benefits of the two? Based on the introduction, it is impossible to tell whether or not this is new research.

(2) Line 49, nutritional improvement of meat products? Do you include nutritional improvement?

(3) Line 56, not welcome, unwelcome. Line 59, not exhibited, it exhibits. Line 61, affectation??

(4) Materials and Methods. What is the transformation of taste substances, in addition to the change in texture? The gas chromatography experiment is essential.

(5) Table 1. The significance marks on all tables are random. In addition, keep the significant figures of the data the same. If the academic norms are not respected, the paper will be rejected.

(6) I prefer to suggest a two-factor ANOVA, since variable one is the processing method. The second variable is storage time.

Author Response

Response to reviewer 1

We appreciate all the reviewer's kind comments to improve the paper. The modifications introduced according to each of your comments are detailed below and marked in red color in the manuscript.

(1) The author attempts to demonstrate why Pleurotus djamor and Hibiscus byproduct may be used to enhance the quality of meat products. The introduction, however, is inadequate. The author did not explain why both ingredients were added to the meat pie at once. Does synergy exist? What are the combined benefits of the two? Based on the introduction, it is impossible to tell whether or not this is new research.

Response (R): Thank you for your kind comment. We have included some additional sentences (in red color) to make clearer the objective of the paper.

(2) Line 49, nutritional improvement of meat products? Do you include nutritional improvement? 

R: The sentence has been rewritten (in red color) citing a review about the use of different mushrooms to get healthier meat products. In this case, the addition of mushroom provides fiber to the meat product and the replacement of meat by Pleurotus djamor did not modify the protein levels, hardly modifying the amino acid composition. In this sense, we considered that nutritional improvement of the meat product has occurred.

(3) Line 56, not welcome, unwelcome. Line 59, not exhibited, it exhibits. Line 61, affectation?? 

R: Thank you for the grammar corrections. They have been done. At the request from the publisher, an English style and grammar revision of the language was done.

(4) Materials and Methods. What is the transformation of taste substances, in addition to the change in texture? The gas chromatography experiment is essential.

R: We agree that the modification of taste because of the inclusion of mushroom and roselle is a fact. Moreover, the thermal process applied during patties cooking also modifies the flavor of the ingredients, especially when mushrooms are present. There are several works about the modification of mushroom aroma when different processes are applied to mushrooms as well as the increase of acidity when roselle is included (Sun et al., 2020; Zhao et al., 2020; Perez-Baez et al., 2020). In this case, giving the complexity of the topic we considered only to assess the flavor changes by means of a sensorial panel, although the study of taste modifications should be the next step in our research once the most appropriate concentrations are defined.

- Perez-Baez, A. J., Camou, J. P., Valenzuela-Melendres, M., Lucas-Gonzalez, R., Viuda-Martos, M. Assessment of Chemical, Physico-Chemical and Sensorial Properties of Frankfurter-Type Sausages Added with Roselle (Hibiscus sabdariffa L.), Extracts. Proceedings 202070(1), 73. https://doi.org/10.3390/foods_2020-07690

- Sun, L. B., Zhang, Z. Y., Xin, G., Sun, B. X., Bao, X. J., Wei, Y. Y., ... Xu, H. R. Advances in umami taste and aroma of edible mushrooms. Trends in Food Sci. Technol. 2020, 96, 176-187. https://doi.org/10.1016/j.tifs.2019.12.018

- Zhao, X., Wei, Y., Gong, X., Xu, H., & Xin, G. Evaluation of umami taste components of mushroom (Suillus granulatus) of different grades prepared by different drying methods. Food Sci. Hum. Wellness 20209(2), 192–198.

https://doi.org/10.1016/j.fshw.2020.03.003

 (5) Table 1. The significance marks on all tables are random. In addition, keep the significant figures of the data the same. If the academic norms are not respected, the paper will be rejected.

R: Thanks for your warning. Changes suggested by the reviewer have been made in the manuscript.

(6) I prefer to suggest a two-factor ANOVA, since variable one is the processing method. The second variable is storage time.

R: Thank you for the advice, the statement has been amended.

Reviewer 2 Report

Materials and Methods:

The drying  at 40 °C for long time, 2-3 days in an industrial oven will affect the nutritional value or chemical composition?

- Using Hs in preparation of beef burgers can be used as a functional food?

- Is it possible to be used for all people or there is a chance for interference  with some drugs used for hypertension

Author Response

Response to reviewer 2

We appreciate the reviewer’s comments.

Materials and Methods:

(1)- The drying at 40 °C for long time, 2-3 days in an industrial oven will affect the nutritional value or chemical composition?

Response (R): Thank you for your comment. The drying process always involves some modification of the molecules of the foods. However, given the short shelf-life of mushrooms, drying ensures to have a stable ingredient with higher concentrations of nutrients. We think the macromolecules like protein, carbohydrates, as well as mineral contents are not affected, only they are concentrated. However, bioactive compounds like polyphenols or vitamins could be affected. In previous works, we dried at 60 °C, and antioxidant properties or antimicrobial properties were not observed. In this work, we dried at a lower temperature in order to preserve these bioactive compounds, but we needed longer periods to have moisture lower than 10%. An antimicrobial effect was observed but not an antioxidant effect, unlike other works have reported. Lyophilization would be a better option but it is more difficult to implement for small mushroom producers units.

(2)- Using Hs in preparation of beef burgers can be used as a functional food?

R: There are several works on the bioactive compounds of Hs (Banwo et al., 2022; Nayak et al., 2022). However, in this case, we are working with the by-products. Although by-products still have antioxidant properties as shown in Table 2, more studies should be necessary to prove their functional properties on human health.

- Banwo, K., Sanni, A., Sarkar, D., Ale, O., Shetty, K. (2022). Phenolics-Linked Antioxidant and Anti-hyperglycemic Properties of Edible Roselle (Hibiscus sabdariffa Linn.) Calyces Targeting Type 2 Diabetes Nutraceutical Benefits in vitroFrontiers in Sustainable Food Systems 2022. https://doi.org/10.3389/fsufs.2022.660831

- Nayak, P. K., Sundarsingh, A. In vitro gastrointestinal digestion studies on total phenols, flavonoids, anti-oxidant activity and vitamin C in freeze-dried vegetable powders. J. Food Sci. Technol. 2022, 59(11), 4253-4261.

(3)- Is it possible to be used for all people or there is a chance for interference with some drugs used for hypertension

R: We would like to have the answer, but we have not studied the relationship of the inclusion of Pd or Hs with specific illness treatments. However, since mushrooms or Roselle drinks are not contraindicated in hypertensive patients, it should not be a problem. We have even found some studies about roselle drinks to reduce hypertension. But as we said, this should be studied.

Reviewer 3 Report

This paper describes the effects of Hibiscus sabdariffa (roselle) and Pleurotus diamor on the physicochemical and sensorial properties of low-salt beef patties. The manuscript is well developed, the objectives are clear.  The results support the conclusions.

Title

-        the expression “to improve low-salt beef patties” is not clear and is not entirely adequate to the content of publication

Materials and methods:

-        the methods are mostly appriopriate

Results:

-        the results are well discussed

-        number of bacteria - in the methodology is cfu/g but in the results and Figures is ufc/g?   

Figures (2 and 3) are not very clear and should be improved. I recommended modify figures to distinguish study variants in a better way.

PCA analysis is recommended to determine the interrelation between quality attributes and samples.

References:

-        the selection of literature and the presentation of literature data is appriopriate,

but the manner of presentation of literature data should be uniform, there are a lot of mistakes

-        names of bacteria and plants should be written in italics (l. 648; l. 674; …….)

Author Response

Response to reviewer 3

We truly appreciate the reviewer comments to enrich the manuscript. We have tried to address the suggestions as far as possible.

This paper describes the effects of Hibiscus sabdariffa (roselle) and Pleurotus diamor on the physicochemical and sensorial properties of low-salt beef patties. The manuscript is well developed, the objectives are clear.  The results support the conclusions.

Title

  • the expression “to improve low-salt beef patties” is not clear and is not entirely adequate to the content of publication

Response (R): Thank you for your comment. The title has been modified accordingly.

Materials and methods:

  • the methods are mostly appriopriate

R: Thank you, we have tried to better explain in order to make the experiments reproducible

Results:

  • the results are well discussed

R: Thank you so much for your comment.

-   number of bacteria - in the methodology is cfu/g but in the results and Figures is ufc/g?   

R: Thank you so much for your observation. The units have been corrected in the graphs and text (changes are in red).

Figures (2 and 3) are not very clear and should be improved. I recommended modify figures to distinguish study variants in a better way. 

R: Thank you for your suggestion. Following the reviewer's suggestion, Figure 2 has been modified and Figure 3 has been converted into a table to facilitate the understanding of the results.

PCA analysis is recommended to determine the interrelation between quality attributes and samples.

R: We appreciate your suggestion. In this case we do not consider the PCA analysis because it did not provide more substantial information than the reported in the text.

References:

-   the selection of literature and the presentation of literature data is appriopriate, 

but the manner of presentation of literature data should be uniform, there are a lot of mistakes

names of bacteria and plants should be written in italics (l. 648; l. 674; …….)

R: Thank you for your kind comment. We used Mendeley reference manager software, but it has some errors like the absence of italics in microorganisms. We have carefully checked the references section.

Reviewer 4 Report

Foods

Foods-2087655

Hibiscus sabdariffa by-product and Pleurotus djamor to improve low-salt beef patties

Dear Editor,

The article deals with the determination of the effect of the addition of Hibiscus sabdariffa (roselle) by-product and Pleurotus djamor (pink oyster) powder as meat replacers on the physicochemical and sensorial properties of low-salt beef patties. The topic is good. However, it lacks some important issues.

-       Line 79 and throughout the ms: Please give g values not rpm for centrifugation process!

-       Line 85: Why did the researchers use methanolic solution do?

-       Line 120: What is the reason for not performing chemical composition and salt analyses during storage?

-       Lines 125 and 126: What about non-protein nitrogenous susbtances?

-       Throughout the ms: “enterobacteria” or “Enterobacteriaceae”?

-       “Moisture” should be “water” throughout the ms due to the values of the samples were higher than 50%.

-       How did the researchers choose the groups?

-       How many repetitions were applied for the analyzes?

-       Did the researchers analyze the meat (pH, water content, TBARS etc.) they used as the material?

-       Has the amount of meat in the meatball portion been calculated? What if the results obtained (eg TBARS, cooking loss etc.) are related to a decrease in the amount of meat in the meatballs? As a result, as the % of slice used increases, the amount of meat in the meatballs will decrease.

-       What is the reason for the changes in the a* value in the control group samples? Myoglobin, specifically iron oxidation? If so, why the value decreased, increased and then decreased again?

-       Were the meatballs within consumable limits in terms of microbiological analysis?

Author Response

Response to reviewer 4

The authors thank the reviewer´s effort and the valuable comments given.

The article deals with the determination of the effect of the addition of Hibiscus sabdariffa (roselle) by-product and Pleurotus djamor (pink oyster) powder as meat replacers on the physicochemical and sensorial properties of low-salt beef patties. The topic is good. However, it lacks some important issues.

  • Line 79 and throughout the ms: Please give g values not rpm for centrifugation process!

Response (R): We thank the observation and the units have been modified according to your suggestion. Changes have been marked in red color.

  • Line 85: Why did the researchers use methanolic solution do?

R: We think that the reviewer is referring to the antioxidant activity determination of Pd and Hs powders. Several solvents can be used in the extraction of antioxidant compounds in vegetables, being the most common water, methanol, acetone, etc., and their combination. The extraction will depend on the polarity of the antioxidant compounds, which is usually related to the presence of polyphenols. Methanol has been reported as a solvent for the extraction of antioxidants in several works and we decided to use this solvent.

  • Line 120: What is the reason for not performing chemical composition and salt analyses during storage?

R: During the storage, we assumed that chemical composition hardly or minimally changed, since spoilage was focused on microbial growth. Despite proteolysis occurring during storage, the measure of protein is based on nitrogen quantification (Kjeldahl method) and it does not vary during storage. In addition, the salt content does not change during storage.

  • Lines 125 and 126: What about non-protein nitrogenous substances?

R: We are not sure about understanding this question. Protein is calculated from the nitrogen determination using specific protein conversion factors for each group of foods and avoiding the non-protein nitrogenous substances. The most common factor used is 6.25, but for example, 4.38 was used in the calculation of Pleurotus djamor protein content.

  • Throughout the ms: “enterobacteria” or “Enterobacteriaceae”?

R: Thank you for your suggestion. The text has been revised to use enterobacteria as the common name of the species belonging to the Enterobacteriaceae family.

  • “Moisture” should be “water” throughout the ms due to the values of the samples were higher than 50%.

R: Thank you for your suggestion. The revised literature consulted for this experiment, including other studies focused on the addition of several ingredients in patties, considers the term moisture in foods independently of the water concentration and we followed this standard.

  • How did the researchers choose the groups?

R: We are very sorry but at this point, we do not understand if the reviewer is referring to the way we select the panelists. The people who performed the sensory analysis, both sexes and aged between 29-45 years, were selected based on their interest to participate and their availability for the evaluation. If the reviewer is referring to how formulations were selected, we did preliminary experiments on the maximum amount of Pd and Hs that we could consider without a great impact on sensory properties and obtaining good nutritional properties.

  • How many repetitions were applied for the analyzes?

R: On each sampling day, two packs were analyzed independently and each analysis was done in triplicate.

  • Did the researchers analyze the meat (pH, water content, TBARS etc.) they used as the material?

R: The original meat was not analyzed but the control samples were only the meat added with 0.7% of salt and 10% of water. So, values obtained for control samples could be considered representative of the original meat, except for the salt and moisture concentration. 

  • Has the amount of meat in the meatball portion been calculated? What if the results obtained (eg TBARS, cooking loss etc.) are related to a decrease in the amount of meat in the meatballs? As a result, as the % of slice used increases, the amount of meat in the meatballs will decrease.

R: The amount of meat in the patty was not calculated. However, we think the percentage of meat replaced was low and the differences seem to be more related to the ingredient added. For example, in Pd5 and Hs5 samples, the same 5% of the meat was replaced by Pd or Hs, but the results in TBARS, cooking loss, textural properties, etc., were very different. This can be only explained by the added ingredient, not by reducing the meat proportion.

  • What is the reason for the changes in the a* value in the control group samples? Myoglobin, specifically iron oxidation? If so, why the value decreased, increased and then decreased again?

R: Thank you for your comment. According to the literature, the decrease of a* value during cold storage by oxidation of myoglobin is expected. The decrease at the end of storage is clearly observed. The variations reported during 0, 4 and 7 days could be attributed to the heterogeneity of the sample. Several meat patties were employed, and this meat had around 3% of fat, which could affect the color). Finally, the oxidation did not progress uniformly which provoked variations in the measurements.

-       Were the meatballs within consumable limits in terms of microbiological analysis?

R: The beef patties could be consumed on day 0. In fact, they were eaten for the sensory analysis. On day 4, only samples codified as Hs5 and Pd7.5-Hs2.5 presented TVC counts below 6 Log CFU/g, which is the microbial limit considered for consumption, as it has been described in the discussion.

Round 2

Reviewer 4 Report

Dear Editor,

The authors have revised and improved their manuscript according to the reviewers' comments and suggestions. Therefore, the manuscript can be accepted and published in its current form.

Best regards,

Author Response

Thank you for your kind answer.